



# Size segregated particle number and mass emissions in urban Beijing

Jing Cai[1,2], Biwu Chu[1,2,3,4], Lei Yao[2], Chao Yan[1,2], Liine M. Heikkinen[1,2], Feixue Zheng[1], Chang Li[1], Xiaolong Fan[1], Shaojun Zhang[5], Daoyuan Yang[5], Yonghong Wang[2], Tom V. Kokkonen[1,2], Tommy Chan[1,2], Ying Zhou[1], Lubna Dada[1,2], Yongchun Liu[1], Hong He[3,4], Pauli Paasonen[1,2], Joni T. Kujansuu[1,2], Tuukka Petäjä[1,2], Claudia Mohr[6], Juha Kangasluoma[1,2], Federico Bianchi[1,2], Yele Sun[7], Philip L. Croteau[8], Douglas R. Worsnop[2,8], Veli-Matti Kerminen[1,2], Wei Du[1,2]*, Markku Kulmala[1,2]*, Kaspar R. Daellenbach[1,2]*

[1] Aerosol and Haze Laboratory, Beijing Advanced Innovation Center for Soft Matter Science and Engineering, Beijing University of Chemical Technology, Beijing, 100029, China

[2] Institute for Atmospheric and Earth System Research, Faculty of Science, University of Helsinki, Helsinki, 00014, Finland

[3] Center for Excellence in Regional Atmospheric Environment, Institute of Urban Environment, Chinese Academy of Sciences, Xiamen, 361021, China

[4] State Key Joint Laboratory of Environment Simulation and Pollution Control, Research Center for Eco-Environmental Sciences, Chinese Academy of Sciences, Beijing, 100085, China

[5] School of Environment, Tsinghua University, Beijing, 100084, China

[6] Department of Environmental Science, Stockholm University, Stockholm, 11418, Sweden

[7] State Key Laboratory of Atmospheric Boundary Layer Physics and Atmospheric Chemistry, Institute of Atmospheric Physics, Chinese Academy of Sciences, Beijing, 100029, China

[8] Aerodyne Research, Inc., Billerica, MA, 01821, USA

*Correspondence to:* Wei Du (wei.du@helsinki.fi), Markku Kulmala (markku.kulmala@helsinki.fi) and Kaspar R. Daellenbach (kaspar.daellenbach@gmail.com)

**Abstract.** Although secondary particulate matter is reported to be the main contributor of $PM_{2.5}$ during haze in Chinese megacities, primary particle emissions also affect particle concentrations. In order to improve estimates of the contribution of primary sources to the particle number and mass concentrations, we performed source apportionment analyses using both chemical fingerprints and particle size distributions measured at the same site in urban Beijing from April to July 2018. Both methods resolved factors related to primary emissions, including vehicular emissions and cooking emissions, which together make up 76% and 24% of total particle number and organic aerosol (OA) mass, respectively. Similar source-types, including particles related to vehicular emissions ($1.6\pm1.1$ µg m$^{-3}$; $2.4\pm1.8\times10^3$ cm$^{-3}$ and $5.5\pm2.8\times10^3$ cm$^{-3}$ for two traffic-related components), cooking emissions ($2.6\pm1.9$ µg m$^{-3}$ and $5.5\pm3.3\times10^3$ cm$^{-3}$) and secondary aerosols ($51\pm41$ µg m$^{-3}$ and $4.2\pm3.0\times10^3$ cm$^{-3}$) were resolved by both methods. Converted mass concentrations from particle size distributions components were comparable with those from chemical fingerprints. Size distribution source apportionment separated vehicular emissions into a component with a mode diameter of 20 nm ("Traffic-ultrafine") and a component with a mode diameter of 100 nm ("Traffic-fine"). Consistent with similar day and night-time diesel vehicle $PM_{2.5}$ emissions estimated for the Beijing area, Traffic-fine, hydrocarbon-like OA (HOA, traffic-related factor resulting from source apportionment using chemical fingerprints), and black carbon (BC) showed similar diurnal patterns, with higher concentrations during the night and morning than





during the afternoon when the boundary layer is higher. Traffic-ultrafine particles showed the highest concentrations during the rush-hour period, suggesting a prominent role of local gasoline vehicle emissions. In the absence of new-particle formation, our results show that vehicular related emissions (14% and 30% for ultrafine and fine particles, respectively) and cooking activity related emissions (32%) dominate the particle

number concentration while secondary particulate matter (over 80%) governs $PM_{2.5}$ mass during the non-heating season in Beijing.

### 1. Introduction

Even though it is commonly recognized that secondary aerosol mass governs haze formation in megacities in China (Huang et al., 2014;Zhang et al., 2013;Tao et al., 2017;Sun et al., 2018), the contributions of primary

(direct) particle sources cannot be neglected. Previous studies have demonstrated that primary emission sources, such as residential heating, traffic and cooking activities, can contribute significantly to both particle number and mass concentrations in urban atmosphere in China (He et al., 2004a;Xu et al., 2014;Du et al., 2017;Wang et al., 2013;Sun et al., 2018). It was recently reported that traffic could be a major source of nanoclusters (sub-3 nm) in urban environments (Ronkko et al., 2017). On average, 13–24% of the total fine

organic aerosol mass concentration (OA) can be attributed to cooking activities and 11–20% to traffic emissions in Beijing, China (Hu et al., 2016;Hu et al., 2017).Together with direct particle emissions, many identified primary sources co-emit high concentrations of volatile organic compounds (VOCs), which in turn contribute to secondary organic aerosol mass formation (SOA) (Liu et al., 2017a;Liu et al., 2017b). Therefore, it is important to identify primary particle sources, and disentangle them from the secondary organic and

inorganic aerosol (SOA and SIA) whose precursors were co-emitted, with the goal to better understand their contributions in highly complex urban atmospheres for advising air pollution control policies.

Beijing, a megacity with a population of 20 million, has suffered from severe fine particulate matter (PM) pollution for several decades (He et al., 2001;Tao et al., 2017). Due to its impact on human health and the climate, fine particulate matter has gained increased attention (Lelieveld et al., 2015;Huang et al., 2014). To

study the fine PM sources in Beijing, numerous receptor source apportionment studies have been conducted (Ding et al., 2016;Tao et al., 2017;Hu et al., 2016;Zhang et al., 2013). These studies can be grouped into two approaches: the widely-applied chemical component method (Xu et al., 2019;Hu et al., 2017;Sun et al., 2013b;Sun et al., 2018) and the less-applied size distribution method (Wang et al., 2013;Liu et al., 2017c;Du et al., 2017), based on the variations of chemical component and size distribution of fine PM from different

sources, respectively.

Most of the aforementioned source apportionment studies in Beijing based on the chemical component method used data from online or offline measurements for receptor models, such as positive matrix factorization (PMF) and chemical mass balance (CMB) (Zhang et al., 2013;Zhang et al., 2017b;Tao et al., 2017). Among them, many focused on the sources of OA due to its large contribution to fine PM, its complex

mix of origins and its tracers from different sources. The development of aerosol mass spectrometer



technologies allowed identification of the primary sources of OA in Beijing, namely traffic emissions, cooking activities, biomass burnings and coal combustions (Xu et al., 2019;Hu et al., 2017;Sun et al., 2013b;Sun et al., 2018). Generally, in OA source apportionment using chemical fingerprints from mass spectrometers (ACSM, AMS), particle size distributions are disregarded. Results of source contributions focus on mass, and not on number, which has however shown to be of importance from a health perspective as well. Aerosol mass spectrometers are blind to particles smaller than ~70 nm (Xu et al., 2017). Apportioning smaller particles to their sources, is therefore crucial for air quality mitigation.

Size-distribution-based source apportionment can provide, though less applied and with more uncertainties, size-segregated particle number concentrations of sources and processes. Until now, size distribution source apportionment has successfully been applied to data from U.S., European and Chinese cities, such as London (Harrison et al., 2011;Beddows and Harrison, 2019), New York (Ogulei et al., 2007), Barcelona (Vu et al., 2015) and Beijing (Wang et al., 2013;Du et al., 2017;Liu et al., 2014). Different sources, such as different types of traffic, cooking, road dust, combustion, re-suspension, secondary sulfate and nitrate have been identified in Beijing by this approach (Du et al., 2017;Liu et al., 2014;Vu et al., 2015;Wang et al., 2013). Yet, until now, very few studies have combined particle number size distribution source apportionment with chemical speciation source apportionment and compared their results in a comprehensive manner. In general, size-distribution based source apportionment results tend to lack validation as well as comparison to other methods, which results in larger uncertainties and a necessity to combine it with chemical speciation source apportionment.

In this study, we aim to better constrain the chemical and physical properties of primary organic aerosol in Beijing using particle number size distribution and chemical speciation source apportionment approaches. We applied both chemical fingerprints (OA-PMF) and particle size distribution (Size-PMF) analyses to resolve the particle mass and number contributions from various sources during the same period. Combining physical information with chemical characterization is crucial for studies on health and atmospheric pollution. In the complex atmosphere of Beijing, we found that on days with no signs of new particle formation (NPF), primary emissions, from for example traffic and cooking, contributed most to the particle number concentration below 100 nm while secondary mass formation dominated the total particle mass concentration.

## 2. Methodology

### 2.1 Measurement site and instrumentation

The sampling site is located in the west campus of Beijing University of Chemical Technology (BUCT, 39° 56'31" N, 116°17'50" E), near the West Third Ring Road of Beijing. The measurement station is located on the top floor of a five-floor teaching building (about 20 m above the ground level). The sampling site is surrounded by residential areas with possible local emissions, such as traffic and commercial and domestic cooking. The station can be viewed as a typical urban residential site in Beijing. Detailed information of the sampling site can be found in Zhou et al. (2020). Our sampling period was from April 6 to July 2, 2018 (84





days), which is outside the heating period (usually mid-November to mid-March). Ambient daily average temperature ranged between 8.2 and 34 °C during the sampling period. Few days were excluded due to necessary calibrations and power cuts (April 27th and June 5th to 13th for size distribution, April 26th –29th June 11th–12th, and 26th–28th for chemical component measurement). In our study, we did not find contributions of

residential coal combustion and biomass burning for cooking and heating (Figure 5 and SI 2.2). Coal and biomass burnings are more important during winter (Hu et al., 2017;Sun et al., 2018). In addition, the transition in energy consumption from coal burning to natural gas and electricity in urban Beijing took place from the year of 2009 to 2017, which led to a decrease in the proportion of coal to total primary energy consumption from 43% in 2007 to less than 20% in 2015 (Zhang et al., 2018).

An online Time-of-Flight-Aerosol Chemical Speciation Monitor (ToF-ACSM, Aerodyne Research Inc.) equipped with a $PM_{2.5}$ lens and standard vaporizer was operated at the BUCT site. A $PM_{2.5}$ cyclone was deployed on the rooftop with a flow rate of 3 L.min$^{-1}$ and connected to the ToF-ACSM by a 3-m stainless steel tube. ACSM and related techniques have been widely used to measure the concentrations of non-refractory (NR) PM components, including sulfate, nitrate, ammonium, organics and chloride, and to identify sources of

fine PM in different environments around the world (Crippa et al., 2014;Huang et al., 2014;Jimenez et al., 2009;Zhang et al., 2011). The black carbon (BC) component of $PM_{2.5}$ was measured by a co-located 7-wavelength aethalometer (AE33, Magee Scientific Corp.) with a sampling flow rate of 1 L.min$^{-1}$. To measure gas phase tracers, a nitrate- Chemical Ionization- Atmospheric Pressure interface- Time of Flight- mass spectrometer (nitrate-CI-APi-TOF (Jokinen et al., 2012), Aerodyne Research Inc.) was also deployed at the

same station from 28th May to 10th June, 2018.

Size-resolved particle number concentrations (size range 20 to 680 nm) were measured with a scanning mobility particle sizer (SMPS, Model 3936, TSI Corp.) at a 5-min time resolution. The SMPS was equipped with a long differential mobility analyzer (LDMA, Model 3080, TSI Corp.) and a condensation particle counter (CPC, Model 3775, TSI Corp.) with a sampling flow rate of 0.3 L. min$^{-1}$, as well as with a $PM_{2.5}$ cyclone. To

reduce sampling losses, an extra pump (flow rate of 16.7 L min$^{-1}$) was used, which resulted in a residence time in the sampling lines shorter than 1.2 s. The atmospheric boundary layer height (BLH) was measured from the optical backscattering of the ceilometer observations (CL-51, Vaisala Inc.) by applying a three-step idealized-profile proposed by Eresmaa et al (2012). Trace gases, including CO, $SO_2$, $NO_x$ and $O_3$, were also measured at the same site (48i, 43i-TLE, 42i, and 49i, Thermo Environmental Instruments Inc.). Total $PM_{2.5}$ and $PM_{10}$

concentrations used in this study were measured by the China National Environmental Monitoring Center (CNEMC) and averaged over the four nearest monitoring stations of: Wanliu, Gucheng, Wanshouxigong and Guanyuan.

In this study, the SMPS and ACSM cover the size ranges of 20 to 680 nm and ~100 to ~2500 nm, respectively (Xu et al., 2017). In addition, the SMPS measures the mobility diameter ($D_m$) of particles, while

the ACSM uses the vacuum aerodynamic diameter ($D_{va}$) for defining the measurement particle size range. Since we used both these instruments for our study, we had to determine the overlapping size range of these two instruments. For spherical particles, $D_{va}$ is assumed to be roughly equal to $D_m$ multiplied by the particle





chemical component density ($\rho_{comp}$) (DeCarlo et al., 2004). According to the calculations of $\rho_{comp}$ shown below, the overlapping $D_{va}$ size range was found to be around 100 – 1000 nm, which is the dominant size range of

PM$_{2.5}$.

We estimated $\rho_{comp}$ using Eq. 1. (Salcedo et al., 2006):

$$\rho_{comp} = \frac{[NO_3]+[SO_4]+[NH_4]+[Cl]+[BC]+[Org]}{\frac{[NO_3]+[SO_4]+[NH_4]}{1.75}+\frac{[Cl]}{1.52}+\frac{[BC]}{1.77}+\frac{[Org]}{1.2}}, \tag{1}$$

where NO$_3$, SO$_4$, NH$_4$, Cl, Org and BC are the hourly concentrations of nitrate, sulfate, ammonium, chloride, organics and BC measured by the ACSM and aethalometer. The densities of ammonium nitrate and ammonium

sulfate, ammonium chloride, organic aerosols and BC were assumed to be equal to 1.75, 1.52, 1.77 and 1.2 g cm$^{-3}$, respectively (Park et al., 2004;Poulain et al., 2014;Turpin and Lim, 2001). Through this method, we calculated time- and chemical-component-dependent densities of particles. The obtained densities were used to convert the $D_m$ diameters from the SMPS measurements into $D_{va}$, and to calculate size-segregated mass concentrations from particle number size distributions. The SMPS and ACSM instruments compared well with

each other when using the calculated densities and assuming spherical particles (Figure S1). The calculated mass concentration from SMPS (PM$_{SMPS}$) agreed well with NR-PM$_{2.5}$ plus BC during non-NPF days, which were used for source apportionment (PM$_{SMPS}$ = 0.90×PM$_{NR-PM2.5+BC}$ +0.26, $r$ = 0.94, Figure 1a). The slope between these two concentrations decreased at high PM$_{2.5}$ levels, which was likely due to an increasing fraction of particles larger than the SMPS size range during the haze periods. Overall, the good comparison suggests a

stable performance of both instruments during the sampling period. We may, therefore, conclude that the overlapping size range of the two instruments was validated. Mineral particles (from e.g. dust storms) are not measured by the ACSM, and likely not by the SMPS (with an upper size cut off at around 700nm and 1000nm $D_m$ and $D_{va}$, respectively) due to their sizes (Zhang et al., 2003).

**2.2 Data treatment and source apportionment analysis**

Non-refractory PM$_{2.5}$ (NR-PM$_{2.5}$), which includes sulfate, nitrate, ammonium, organics and chloride, was obtained by using the standard ToF-ACSM data analysis software (Tofware ver. 2.5.13) within IgorPro ver. 6.3.7.2 (Wavemetrics). The relative ionization efficiencies (RIE) for sulfate, nitrate, ammonium, chloride and organics applied were 0.86, 1.05, 4.0, 1.5 and 1.4, respectively. Among those, the RIEs of ammonium, sulfate and chloride were determined by calibrations with pure standards. The good ion balance during the sampling

period suggests those RIEs to be acceptable (NH$_{4measured}$=1.0×NH$_{4calculated}$ - 0.27, $r$ = 1.0). A chemical component-dependent collection efficiency (CE) was applied to the ACSM data with the correction method suggested by Middlebrook et al. (2012). The sum of NR-PM$_{2.5}$ and BC correlates with PM$_{2.5}$ from the surrounding measurement sites (CNEMC, $r$ = 0.82), except for periods with higher coarse particle concentrations in spring shown by higher ratios of coarse to fine particle mass concentrations (Figure1b),

particularly pronounced during a dust storm in May, 2018. However, the chemically resolved PM$_{2.5}$ (sum of NR-PM$_{2.5}$ and BC) accounts overall for 77% of the bulk PM$_{2.5}$, indicating the presence of refractory PM such as mineral dust.



In this study, positive matrix factorization (PMF) was applied separately to chemical fingerprints of OA from
ToF-ACSM (OA-PMF) and particle number concentrations from SMPS datasets (Size-PMF). For OA-PMF,
the mass spectra of organics were imported into the Source Finder toolkit (SoFi, ver. 6.8.4), which
communicates with the multi-linear engine (ME-2). In this study, the partially constrained method utilizing $a$-
value and pulling equations was applied. Detailed information of SoFi and the $a$-value method are described
elsewhere (Daellenbach et al., 2016;Canonaco et al., 2013).

In Size-PMF, the ambient size-resolved number concentrations measured with an SMPS were imported (total
of 106 size bins) to the PMF model (PMF 2, ver. 4.2) (Ulbrich et al., 2009). To better estimate the uncertainties
in Size-PMF, a typical uncertainty estimation method for size-resolved number concentration data was applied
(Du et al., 2017;Ogulei et al., 2007). The uncertainties were defined as heuristic errors, $\sigma_{ij}$, which were
calculated based on the measurement errors, $S_{ij}$:

$$S_{ij} = C_1 \times (X_{ij} + \overline{X}_J) \qquad (2)$$

$$\sigma_{ij} = S_{ij} + C_2 \times X_{ij} \quad , \qquad (3)$$

where $C_1$ and $C_2$ are constants with values of 0.01 and 0.1, respectively, which are proposed by Ogulei et al.
(2007) according to residual distributions and typically applied in SMPS PMF analyses (Du et al., 2017). $X$ is
the measured particle number concentration in the $j^{th}$ size bin at the $i^{th}$ point in time. $\overline{X}_J$ is the arithmetic average
for the $j^{th}$ size bin. In the PMF analysis, size-resolved particle number concentrations were averaged to a 15-
min resolution and NPF days were excluded to better estimate contributions from the primary sources. NPF
events where classified by the appearance of nucleation mode particles showing signs of growth following the
methods proposed by Dal Maso et al. (2005) and Kulmala et al. (2012). In this study, the classification relied
on the particle number size distribution measured using the SMPS and a complementary NAIS (2-42 nm -
Neutral cluster and Air Ion Spectrometer, (Manninen et al., 2011;Mirme and Mirme, 2013) for confirming our
classification. Of our sampling period, 39 NPF days were excluded and exact dates are provided in Table S1.
Haze days were classified based on visibility. When the visibility went below 10 km while the relative humidity
did not exceed 90% for consecutive 8 hours, the day was defined as a haze day (Zhou et al. 2020).

### 2.3 Traffic emissions

Yang et al. (2019) used real-world traffic observations from major roads in Beijing, including hourly traffic
volume, speed and vehicle mix information, and the real-time congestion index as well as traffic density
modeling to map street-level, hourly traffic data for 2013 and 2017. After several years of traffic monitoring,
no significant changes in diurnal traffic patterns were found across various years or seasons (Song et al.,
2013;Han et al., 2009). This study used the hourly traffic profiles for the nearest arterial road (Zizhuqiao Road),
to represent average traffic conditions close to the monitoring site.

Emissions of $PM_{2.5}$ from different categories of vehicles in urban (within $5^{th}$ ring road) and whole Beijing
area were estimated by using the EMBEV-Link (Link-level Emission factor Model for the BEijing Vehicle
fleet) model (Yang et al., 2019). This model is based on multiple datasets extracted from the extensive road





traffic monitoring network in Beijing, including vehicle speed, traffic volume and fleet types. Speed-dependent
emission factors were also applied to estimate the vehicular $PM_{2.5}$ emissions from different types of vehicles
in urban (within 5[th] ring road) and Beijing area (whole city area). Detailed information of this model and results
can be found in Yang et al. (2019).

## 3. Results and discussion

### 3.1 Time variations of fine PM

The temporal variation of size-binned particle number concentration, chemical components of NR-$PM_{2.5}$,
temperature and RH for all sampling days are displayed in Figure 2 a, b and c. During the sampling period,
frequent new particle formation events occurred, in line with previous studies (Wu et al., 2007;Chu et al.,
2019;Zhou et al., 2020). Organic aerosols, nitrate and ammonium were the dominating contributors to the total
$PM_{2.5}$ mass concentration, especially during the haze events, showing the contribution of secondary species to
the high PM concentration and haze formation. The sharp decreases of the particle number and mass
concentrations in the sampling period can be explained by the arrivals of cold fronts and precipitation. The
cold fronts are indicated by a wind shift from the South to the North and significant increase of wind speed
shown in Figure 2 d. Generally, despite their strong diurnal cycle, the temperature increased from spring to
summer during the sampling period. Moreover, PM concentrations appear in general to be elevated at increased
relative humidity (RH). This can be attributed to air masses with higher humidity transported from the polluted
South, together with heterogeneous secondary PM formation (Cai et al., 2017;Jia et al., 2008).

     Result of the effective air pollution control strategies from the Chinese government in the recent years
(Cheng et al., 2019;Wang et al., 2019b), the average $PM_{2.5}$ during our sampling period was 56±40 µg m$^{-3}$
(53±40 µg m$^{-3}$ for the year of 2018), far lower than the levels from the early 2010s in Beijing (annual average:
135±63 µg m$^{-3}$ for 2013)(Zhang et al., 2013;Sun et al., 2013a;Sun et al., 2013b). The average number
concentrations in the size range from 20 nm to 680 nm were around $1.6\times10^4$ cm$^{-3}$, which is close to the values
observed at other urban and regional sites in China, such as Guangzhou ($1.4\times10^4$ cm$^{-3}$), Shanghai ($1.3\times10^4$ cm$^{-3}$) and Wuxi ($1.8\times10^4$ cm$^{-3}$), and 2–3 times higher than the values observed at background sites (e.g. Wenling
$5.7\times10^3$ cm$^{-3}$ and Changdao $6.7\times10^3$ cm$^{-3}$) or in marine environments ($5.6\times10^3$ cm$^{-3}$) (Peng et al., 2014).

The average size distributions of NPF days, haze days and no NPF nor haze event days are displayed in
Figure 3. During NPF days, sub-30 nm particles concentrations are much higher than during non-NPF days.
During NPF days, an additional shoulder in the particle size distribution can be observed at 50 nm, which is
likely affected by primary particle emissions. Contrarily, during haze days, the fraction of sub-30 nm particles
are much lower while the fraction of large particles strongly increased, especially for particles >200 nm. This
suggests an enhanced contribution of secondary formation and regional transport. The size distribution of
particles during the days without neither NPF nor haze events are in between NPF and haze days.





### 3.2 Characteristics of PM and PMF analyses

The average diurnal evolution of the particle number size distribution, calculated over the non-NPF days, showed a clear impact of the primary emissions around the early noon, evening and midnight (Figure 4a). We further divided the particles into two size groups: particles smaller than 100 nm ($N_{20\text{-}100}$ (20–100 nm)) and particles larger than 100 nm ($N_{100\text{-}680}$ (100–680 nm)) in diameter. In general, $N_{20\text{-}100}$ and $N_{100\text{-}680}$ contributed to 66% and 34% of the total particle number concentration ($N$), respectively. Unlike $N_{20\text{-}100}$, the value of $N_{100\text{-}680}$ kept relatively stable over the course of the day (especially for particles > 200 nm), implying that these large particles were likely of regional origin and were not strongly impacted by local primary emissions. A strong source of particles in the size range of 20–100 nm was observed from 17:00 to 23:00, when also the highest particle number concentrations of the whole day were observed. Although it is expected that on non-NPF days, the highest particle number concentration is observed during the nighttime with the lowest boundary layer conditions, this was not the case in our study (Figure 4a, b – 17:00 to 23:00), which indicates that the particle number concentration is driven not only by the boundary layer height but also by primary sources during 17:00 to 23:00. The variations of the fractions of $N_{20\text{-}100}$ and $N_{100\text{-}680}$ are similar to the aforementioned pattern and displayed in Figure S2.

Organic aerosols and nitrate were the largest contributors to the total PM$_{2.5}$ mass (21.8 μg m$^{-3}$, 35% and 17.9 μg m$^{-3}$, 28%, respectively), followed by sulfate, ammonium, BC and chloride. Unlike secondary inorganic species, the diurnal pattern of organics showed an obvious increase influenced by local primary emissions during the morning, noon and evening at around 07:00, 12:00 and 19:00, respectively. Peaks of the organic aerosol mass concentrations and fractions were found at noon (16.5 μg m$^{-3}$, 34% of NR-PM$_{2.5}$+BC) and evening (24.9 μg m$^{-3}$, 46% of NR-PM$_{2.5}$+BC). On the contrary, secondary inorganic aerosols showed very different diurnal patterns, which suggests that they were affected more by regional/ageing processes and boundary layer effects.

We applied OA-PMF and Size-PMF analysis to classify the major sources during the non-NPF days. The evaluation of the OA-PMF and Size-PMF results and validations can be found in SI Section 2. During our sampling period, the fraction of OA observed at $m/z$ 60 (assumed to be affected by levoglucosan from biomass burning (Elser et al., 2016)) and at $m/z$ 115 (assumed to be affected by PAHs from coal combustion (Li et al., 2017;Hu et al., 2017)) was the lowest in comparison to the whole year of measurements (Figure 5 and Figure S5), which indicates that no substantial biomass burning or coal combustion were taking place during our study period. This result is consistent with observations in similar seasons of previous years (2011–2013) (Hu et al., 2017;Sun et al., 2018). In addition, the slope of $m/z$ 60 to total OA concentration ($f_{60}$, 0.004) is very close to background $f_{60}$ level without biomass burning activities ($f_{60}$, 0.003) proposed by Cubison et al. (2011), confirming that biomass burning emissions were only a minor contributor to OA during our sampling period. In the absence of identifiable OA components related to residential biomass burning or coal combustion, primary OA (POA) can be assumed to be dominated by traffic and cooking emissions in Beijing. By using the ME-2 method to obtain a better extraction of the factors and separation of similar profiles, prior information of the source profiles from hydrocarbon-like organic aerosol (HOA) and cooking organic aerosol (COA) were



applied to constrain the PMF runs (Crippa et al., 2014;Crippa et al., 2013b;Crippa et al., 2013a). These profiles were very similar to other previous source tests (Mohr et al., 2012;He et al., 2010;Eilmann et al., 2011) and can be recognized as the typical profiles of their source types. Due to the high similarity of the mass spectral fingerprints of gasoline and diesel exhausts, OA-PMF is unable to further separate HOA into two factors attributed to those two different types of vehicles even by using gasoline and diesel exhaust fingerprints from

literature as a priori information (Canagaratna et al., 2004;Mohr et al., 2009)(Figure S10). Due to the lower mass contribution and smaller particle size range from gasoline exhausts, HOA from OA-PMF in this study are assumed to be most strongly affected by diesel emissions, which is consistent with previous research (Canagaratna et al., 2004) and further discussed in Section 3.3.

In OA-PMF source apportionment analysis, we separated four factors with distinctly different chemical

composition in OA-PMF: HOA, COA, Less Oxygenated Organic Aerosol (LO-OOA) and More Oxygenated Organic Aerosol (MO-OOA). Here also, the residual analyses showed no noticeable contributions from biomass burning and coal combustions (Figure S12). The contribution of aerosol components to NR-PM$_{2.5}$ and the chemical fingerprints from OA-PMF are displayed in Figure 6. Generally, the source types and contributions were consistent with those from previous studies conducted in the same seasons in urban Beijing

(Hu et al., 2017;Sun et al., 2018).

From the Size-PMF analysis, five factors were resolved based on the particle number size distributions: Traffic-fine particles (Traffic-fine), Traffic-ultrafine particles (Traffic-ultrafine), cooking activity related particles (Cooking-related) and Regional particles. The number and volume size distributions (assuming spherical particles) of the components resolved in Size-PMF are displayed in Figure 7. In the following

sections, we discuss the impact of sources and processes on particle number and mass concentrations.

### 3.3 Traffic related particles

### 3.3.1 Traffic-fine particles

In the OA-PMF, the traffic emissions were represented by a factor dominated by fragments related to hydrocarbon-like OA (HOA, hydrocarbon-like OA), including *m/z* 27, 29, 43, 55, 57 (Figure 6 a). These

fragments and HOA are generally believed to be affected by traffic emissions (especially diesel vehicle emissions (Canagaratna et al., 2004)). In this study, HOA (9% of the total OA) had a high correlation with BC ($r = 0.71$), and the HOA/BC ratio (0.45±0.22) was consistent with traffic emissions (0.4 to 0.79 (Daellenbach et al., 2016)).

From Size-PMF, a corresponding factor was found that correlates with BC and HOA plus BC as well ($r =$

0.55 and $r = 0.65$). This factor is characterized by a single-mode in the number size distribution, with a peak size at around 100 nm (Figure 7c) and termed Traffic-fine. This result supports previous studies that suggested BC and primary organics to be the dominant components in this size range of PM in Beijing (Cruz et al., 2019;Xu et al., 2016;Ding et al., 2019;Su et al., 2018;Sun et al., 2015;Wang et al., 2019a;Hu et al., 2016). Assuming particle densities of 1.5 g cm$^{-3}$ in this size range (Hu et al., 2012;Yin et al., 2015) for estimating the

mass concentration of Traffic-fine, the sum of HOA and BC explains 57% (average: 5.2 μg m$^{-3}$) of the mass



concentrations of traffic-fine from the SMPS (average: 8.2 µg m$^{-3}$) (Figure 8 a, e, and i). The lower mass concentration from the chemically resolved measurements could be related to the following factors: (1) the transmission efficiency of the ACSM PM$_{2.5}$ lens is relatively low for particles that contribute to the volume of Traffic-fine (102–322 nm, 10$^{th}$ and 90$^{th}$ percentile) (Peck et al., 2016); (2) contribution of secondary inorganic compounds to particles emitted from traffic (Zhang et al., 2017a;Xing et al., 2019), influencing the Size-PMF results and thereby resulting in overestimation of Traffic-fine component; (3) there are uncertainties arising from the performance and measurement of the instruments, including SMPS (assumed 31% for PM$_1$, (Buonanno et al., 2009a)), ACSM (assumed ~30% but varied among species, (Budisulistiorini et al., 2014)) and aethalometer (assumed 36% for BC, (Sharma et al., 2017)), as well as from the PMF analyses; and (4) the co-emitted refractory compounds, such as zinc, copper, calcium and phosphate, and brake wear particles which cannot be measured by ACSM, even though these compounds can assumed to be minor (Wright and Institute of Marine, 2000;Dallmann et al., 2014).

Traffic-fine, HOA, and BC showed similar diurnal patterns, with higher concentrations during the night and morning than during the afternoon when the boundary layer is higher (Figure 8a, e and i). We assume the boundary layer to have the biggest influence in our study period, day and night-time diesel vehicle PM$_{2.5}$ emissions are estimated to be similar for the Beijing area. Yet, within urban Beijing (area within the 5$^{th}$ ring road), the PM$_{2.5}$ emissions of diesel vehicles increase during the night, which can be related to the heavy-duty vehicle (HDV) restrictions during day time in Beijing (Song et al., 2013). This suggests that in addition to the emissions in urban Beijing, Traffic-fine, HOA, and BC are also strongly impacted by diesel vehicle emissions from the Beijing area due to their size ranges and life time. Furthermore, Traffic-fine had similar mode number diameters as heavy-duty vehicle emissions from reported source tests or near-road studies in Beijing (Song et al., 2013;Wang et al., 2011;Vu et al., 2015;Wehner et al., 2009). According to previous source tests, typically particles from HDV emissions have the dominant number mode of 40 to 160 nm, which is much larger than those from light-duty vehicles (LDVs, 20 nm or smaller, (Vu et al., 2015)). However, also HDV emissions of some smaller particles (<30 nm) were observed in emission tests and at roadside measurements during the night of Beijing when HDV emissions were dominated (Song et al., 2013). Those small particles from HDVs were not observed in Traffic-fine and explained by another factor.

### 3.3.2 Traffic-ultrafine particles

Ultrafine particles (< 30 nm) were mainly explained by another factor characterized by a large contribution of particles with mode diameter around 20 nm (Figure 8b) to its particle number size distribution. The separated factor exhibits a similar particle number size distribution as gasoline vehicular emissions (both source tests as well as urban roadside measurements) (Wehner et al., 2002;Vu et al., 2015;Du et al., 2017;Liu et al., 2014;Wang et al., 2013). In this study, we minimized the effect of NPF on the Size-PMF results by excluding NPF days based on an evaluation starting from 2 nm particle concentrations from the NAIS. Therefore, we termed this factor Traffic-ultrafine. By assuming a density of 1.5 g cm$^{-3}$, which was typically reported for this





size range in Chinese megacities (Hu et al., 2012;Yin et al., 2015;Qiao et al., 2018), the number concentrations of this component were converted to mass concentrations.

Traffic-ultrafine concentrations were higher during daytime than nighttime, which suggests a prominent impact of local gasoline vehicle emissions (Figure 8b). In the morning, Traffic-ultrafine particle concentrations 365 started to increase concurrent with the morning rush-hour and reached the first peak of the day ($1.8 \times 10^3$ cm$^{-3}$, 0.72 μg m$^{-3}$), which is 1.5 times the background concentration between midnight and 4 am, consistent with enhanced estimated gasoline vehicles emissions in urban Beijing and NO$_x$ concentrations. After that, the concentrations of this component decreased, consistent with the distinct decrease in gasoline PM$_{2.5}$ emissions (Figure 8f). The background late night particle concentrations can be attributed to nighttime cluster formation, 370 or to sub-30 nm particle emissions from HDV emissions (Song et al., 2013;Wehner et al., 2009). Enhanced concentrations between 10:00 and 11:00 could be explained by prevailing winds from North and East (winds from North and East observed on two thirds of all days during this time window). This causes the measurement location to be strongly affected by the main intersections and arterial roads upwind (Figure S15). This is supported by a shoulder of the NO$_x$ peak at the same time of day that was still observed even though NO$_x$ is 375 depleted by increasing concentrations of O$_3$. Bootstrap analyses also confirmed that this peak occurred at this time of day during most of the sampling period.

In the evening (18:00–20:00), the concentration ($2.2 \times 10^3$ cm$^{-3}$, 1.0 μg m$^{-3}$) reached ~2 times the background value ($1.4 \times 10^3$ cm$^{-3}$, 0.55 μg m$^{-3}$), which can be attributed to a decreasing boundary layer height and the evening rush hour peak from gasoline emissions. We note that the evening rush-hour concentration of Traffic-380 ultrafine was 30% larger than the morning rush-hour, which might be related to differences in the ultrafine particle emission factors (EFs) in different traffic regimes. Ultrafine particle (UFP, <100 nm) EFs were e.g. found to be 2 to 4 times higher during traffic congestion than during free flow (Zhai et al., 2016). Yet, additional processes not resolved in our analysis might also contribute to this observation, such as the variation of boundary layer height. The diurnal pattern of the Traffic-ultrafine factor was similar to those of typical 385 UFPs reported in previous near-road measurement (Du et al., 2017;Liu et al., 2014;Wang et al., 2013). In contrast to traffic-ultrafine, NO$_x$ did not show an evening rush-hour peak, which is hypothesized to be related to a prominent daytime photochemical depletion of NO$_x$, validated in Beijing winter in recent research (Lu et al., 2019). In winter when photochemistry is weaker, NO$_x$ showed a clear evening rush-hour peak (Figure S14). Additionally, increased HDV traffic volume during the night could also emit large amounts of NO$_x$ (Tan et al., 390 2019;Dallmann et al., 2013).

We did not observe a similar component in OA-PMF, which is possibly related to the following reasons: (1) small size of the particles not transmitted through the PM$_{2.5}$ ACSM inlet, (2) the low mass concentration of this factor, (3) the difficulties in the separation of this factor from other vehicle types using the chemical fingerprints method. Therefore, we cannot comment on the chemical composition of these particles, a feature 395 which should be addressed in future research.





### 3.4 Cooking activity related particles

We identified a factor related to cooking emissions with a mono-modal number size distribution (20–200 nm, geometric mean diameter (GMD) around 50 nm), contributing 32% to the total particle number concentration (Figure 8c and 8g). This mode is similar to that from cooking emission tests (Li et al., 1993;Yeung and To, 2008;Zhao and Zhao, 2018;Abdullahi et al., 2013;Hussein et al., 2006;Buonanno et al., 2009b) and field observations (Harrison et al., 2011;Du et al., 2017). Typically, Chinese cooking activities release particles with a size range from 30 to 100 nm at typical cooking temperatures (Li et al., 1993;2008;Zhao and Zhao, 2018;2011). The time series of this factor could be explained very well by cooking times.

Consistently, cooking emissions in OA-PMF (COA) are represented by a factor with a chemical fingerprint that is somewhat similar to HOA – with the latter having a much higher 57 to 55 $m/z$ ratio in comparison to that of COA. The 57 to 55 $m/z$ ratio is typically used as the tracer of COA (Daellenbach et al., 2017;Mohr et al., 2012;Mohr et al., 2009). Cooking activities can emit saturated alkanes, alkenes, cycloalkanes and oxygenated species such as organic acids, resulting in a higher intensity of the $m/z$ 55 component (mainly $C_3H_3O^+$ and $C_4H_7^+$) compared to the prominent peak at $m/z$ 57 for HOA, which is the characteristic of saturated hydrocarbons (Mohr et al., 2012). Similar to the typical cooking aerosols estimated from ACSM or AMS studies in Beijing (Ding et al., 2016;Sun et al., 2013b;Hu et al., 2017;Hu et al., 2016), the diurnal patterns of COA in this study exhibited two peaks: one during the lunchtime (11:00 to 12:00, 3.7 μg m$^{-3}$) and the other one during dinnertime (18:00 to 20:00, 6.9 μg m$^{-3}$). The average mass concentration of COA in this study was 2.6 μg m$^{-3}$ (15% of OA and 4% of PM$_{2.5}$). These concentrations are similar to those reported in other studies in Beijing (He et al., 2019;Hu et al., 2017;Hu et al., 2016), but higher than those reported in Europe and the United States (He et al., 2019). The larger amount of UFPs and higher PM concentrations related to cooking emissions in Beijing might be attributed to the much larger magnitude of cooking activities as well as higher cooking temperatures, high fatty ingredients and stir-fry cooking styles that are common in Chinese cooking (Buonanno et al., 2009b;Zhao and Zhao, 2018). To confirm our observations, we used measurements of gas phase tracers from NO$_3$-CIMS. Our measurements show that gas-phase pyroglutamic acid (C$_5$H$_7$NO$_3$), previously identified as a cooking marker in the particle phase (Reyes-Villegas et al., 2018), correlated highly with the COA concentration ($r$ = 0.81, 14 daily averages) in a daily comparison (Figure S18 b). Linoleic acid (C$_{18}$H$_{32}$O$_2$), which is a typical tracer for Chinese cooking (He et al., 2004b;He et al., 2004a;Xu et al., 2018;Schauer et al., 2002;Rogge et al., 1991;Reyes-Villegas et al., 2018;Abdullahi et al., 2013), exhibited a weaker correlation with the COA concentration, likely due to potential interferences from other high signal peaks in the same unit mass from photochemistry during the summer in Beijing (Figure S17 a, b).

The number concentration of Cooking-related particles from Size-PMF exhibited very similar diurnal patterns with COA (Figure 8c), both having a larger dinnertime peak (10.2×10$^3$ cm$^{-3}$) and a lower lunchtime peak (6.7×10$^3$ cm$^{-3}$). Since fatty acids are the major components of Chinese cooking emissions (He et al., 2004a;Wu et al., 2015;Reyes-Villegas et al., 2018), the number concentrations of this emission type could be further converted into mass concentrations by assuming a cooking particle density close to oil (0.85 g cm$^{-3}$) (Reyes-Villegas et al., 2018). A high correlation ($r$ = 0.64, Figure 8o) between COA from ACSM and Cooking


particles from Size-PMF suggests that the results from the two methods are robust. Yet, the concentration of COA was 2.3 times the mass concentration derived from Size-PMF. This is consistent with the previous studies

that found a higher relative ionization efficiency (RIE) of COA ($RIE_{COA}$) compared with other OA (Figure S16) (Xu et al., 2018;Canagaratna et al., 2007). However, additional uncertainties in this comparison may arise from the following aspects: (1) there are differences in the measured size ranges of these two instruments and uncertainties in the density of oil for cooking particles; (2) both PMF analyses have their own uncertainties, especially pertaining to the SMPS without the use of unique cooking tracers; (3) during lunchtime and

dinnertime, the contribution of COA to the NR-PM$_{2.5}$ could significantly increase, making the actual CE higher than the only ammonium nitrate-dependent CE applied in this study.

### 3.5 Secondary/regional effects

In this study, we quantify the impact of primary particle emissions on air pollution in a megacity while accounting for secondary/regional effects using two different approaches. Both Size-PMF and OA-PMF

approaches extracted two separate secondary or regional factors related to the different stages of ageing process in the atmosphere and precursor emissions. The two factors resolved from Size-PMF were related to regional secondary aerosols: Regional 1 and 2, both displaying a bimodal number size distribution with a dominant peak at 200 nm and 400 nm, respectively (Figure 7d and 7e). A much smaller mode in the small size range could be found in both factors, which might be due to transport of small particles or the uncertainties in the

Size-PMF analysis. Regional 2 factor had a larger median diameter, indicating that this factor might have been subject to a longer ageing process than Regional 1. Meanwhile, from OA-PMF, two oxygenated organic aerosol factors are related to Secondary organic aerosol (SOA), less oxidized and more oxidized oxygenated OA (LO-OOA and MO-OOA). They both were dominated by oxygenated fragments (e.g. $m/z$ 44 attributed to $CO_2^+$) and much lower hydrocarbon fragments at the same time. Yet, LO-OOA profile had higher signals of

less oxygenated fragments (e.g. $m/z$ 26 and $m/z$ 43) and much stronger signals of ions with $m/z$ larger than 100. MO-OOA had even lower hydrocarbon fragments and less high-weight fragments, suggesting longer ageing and oxidation processes. A strong correlation was observed between MO-OOA and sulfate ($r = 0.75$), which was another indication of the long ageing processes for this factor. The correlation between LO-OOA and nitrate was lower ($r = 0.64$), which is a feature commonly found in many recent studies in Beijing (Sun et al.,

2018;Zhao et al., 2017). This phenomenon was previously explained by the different formation pathways between nitrate and secondary organic aerosols (Sun et al., 2018).

In total, the two regional factors from Size-PMF contributed to a total of 24% (Regional 1 (19%) and Regional 2 (5%)) to the total particle number concentration, yet are the major contributors to the particles larger than 100 nm (57 %) (Figure 7f). Meanwhile, secondary organic and inorganic aerosols (NO$_3$, SO$_4$, NH$_4$,

Chl) dominated the fine PM mass concentration (85 %, Figure 8h). Here, we compared the sum of secondary organic (LO-OOA, MO-OOA) and inorganic particulate matter with the mass concentration derived for the sum of both Regional factors from Size-PMF (termed Regional, assuming a density of 1.5 g cm$^{-3}$, which is close to the density of secondary aerosols and also to the density of particles in this size range reported



previously (Hu et al., 2012)). Secondary particulate matter and Regional mass concentrations agreed well with
each other ($r = 0.93$, slope=1.1), implying that particles from secondary sources showed more aged signals in
the ACSM and exhibited larger sizes in the SMPS. The diurnal patterns of the secondary sources were likely
driven by boundary layer effects, photochemistry and long range transport (Figure 8d).

### 3.6 Number, surface areas and volume distributions of different sources

To compare the correlations of different factors extracted from OA-PMF and Size-PMF, Pearson correlation
coefficients ($r$) between different factors, BC and $NO_x$ are displayed in Figure 9. HOA from OA-PMF was
found to be strongly correlated with BC, Traffic-fine and $NO_x$, indicating that these components were affected
by a similar source type, very likely HDVs. COA from OA-PMF was correlated best with Cooking-related
particles from Size-PMF. A weak correlation was observed between COA and traffic-ultrafine particles, which
is probably due to both of them being daytime activities. The COA component had no correlation with other
source types and tracers, indicating the robustness of the PMF results by the two approaches. Secondary
aerosols were strongly correlated with the Regional component extracted from Size-PMF, suggesting
secondary PM consists of larger particles (>100nm) of more regional character. Besides, the Secondary
component was also correlated with BC and Traffic-fine, which is consistent with the fact that particles emitted
from HDVs, such as BC particles, may have a longer life time and thus are transported further.

The average particle number, surface area and volume size distributions of different factors are displayed in
Figure 10 (assuming spherical particles). Traffic-ultrafine, Traffic-fine, Cooking, Regional-1 and Regional-2
contributed 14%, 30%, 32%, 19% and 5% to the particle number concentration, respectively (20%, 49%, 27%,
3%, 0.6% of particles with diameters 20-100 nm and 4%, 5%, 35%, 45%, 11% of particles with diameters 100
nm). The number fraction of Traffic-ultrafine in this study is lower than that found in a previous Size-PMF
study using a twin differential mobility particle sizer (TDMPS, 25%) in Beijing (Wang et al., 2013). A potential
reason for this is that we excluded sub-20 nm particles in the Size-PMF analysis. Such particles, originating
mainly from traffic during non-NPF days, can be an important contributor to the total particle number
concentration (Kontkanen et al., 2020).

Although the Traffic-ultrafine, Cooking and Traffic-fine factors contributed the majority (76%) of the total
particle number concentration, they only contributed 28% and 16% to the total surface area and volume
concentration, respectively. Meanwhile, the two regional sources dominated the total surface area (Regional
1: 41%; Regional 2: 31%) and volume (Regional 1: 38%; Regional 2: 47 %) concentrations. This observation
is in agreement with previous findings that the mass concentrations of fine PM are dominated by secondary
mass formation (Huang et al., 2014;Sun et al., 2012;Sun et al., 2013a;Sun et al., 2013b;Guo et al., 2014). Our
results highlight the importance of secondary formation for $PM_{2.5}$ mass as well as the importance of the
contribution of primary sources to the UFP number concentration during the non-heating season in Beijing in
the absence of NPF.



### 3.7 Uncertainties and limitations

Even though promising results and good comparisons were obtained by Size and OA-PMF analyses, there
were some uncertainties and limitations in this study. First, since the size ranges of ACSM ($D_{va}$ ~100 nm – 2.5
µm) and SMPS ($D_{va}$ ~30 nm- 1 µm) do not fully overlap and since the smallest particles cannot be measured
with the ACSM due to its lower detection efficiency at smaller sizes, we could not characterize the chemical
composition of the Traffic-ultrafine particle type. Secondly, we used assumptions, including spherical particles
and component-depended densities, which may create uncertainties especially due to the complexity of
morphological characteristics for PM in Beijing (Li et al., 2011). Regardless of the aforementioned possible
uncertainties, measured $PM_{2.5}$ and $PM_{2.5}$ estimated from SMPS agreed well with each other, suggesting no
strong impacts due to these uncertainties. Finally, we cannot exclude that some cluster formation (even though
NPF days were excluded in this study), secondary reactions or air mass transport could have affected measured
size distributions and thus the results of Size-PMF. Yet, ultrafine particles have a short lifetime and are strongly
affected by local emissions, so their contribution to our identified factors must be minimal especially when
excluding NPF event days.

In this study, we only used data from the non-heating period when no significant biomass burning and coal
combustion activities could be identified. The performance of this method when applied to measurements
affected by a more complex mixture of sources, such as winter time, needs to be assessed in future studies. In
addition, a comparison between the sources among different types of measurement sites needs to be quantified
in future studies.

### 4. Conclusions

We measured the aerosol chemical composition as well as size distribution at an urban site in Beijing
between April and July 2018. By conducting two independent PMF analyses (OA-PMF, Size-PMF), similar
sources, contributions as well as diurnal patterns from primary emissions were extracted, validating the
robustness of the Size-PMF in our study period. Furthermore, we identified and quantified the sources and
processes contributing to the particle number, surface area and mass concentrations. For primary emissions,
Size-PMF extracted both Traffic-fine and Traffic-ultrafine factors, while HOA from ACSM was mainly
influenced by the contribution from HDV emissions. The combination of particle chemical composition and
size shows clearly that during non-NPF days, the aerosol number concentration is dominated by direct
anthropogenic particle emissions while the PM mass concentration is governed by secondary processes. The
methods and results of this study could pave the way for a more comprehensive understanding of primary
sources in Beijing, including the particle size, number and mass. Notwithstanding, it has the potential to
provide detailed physical and chemical characteristics of sources for future studies.


*Data availability*



Data is available from the authors upon request.

*Author Contributions*

JC, WD and KRD designed the research and analyzed the data. JC, WD, KRD, BC, LY, CY, LH, FZ, CL, XF, YW, TK, TC and YZ performed the measurements for this study. SZ, DY provided traffic flow and emission data. JC, WD, KRD, BC, LY, YL, LD, HH, PP, YK, TP, CM, JK, FB, YS, PC, DW, VK and MK interpreted the results and revised the manuscript. MK supported and supervised this research. JC wrote the manuscript with contributions from all co-authors.

All authors have given approval to the final version of this manuscript.

*Competing interests.* The authors declare that they have no conflict of interest.

*Acknowledgements*

The work is supported by Academy of Finland (Center of Excellence in Atmospheric Sciences, project no. 307331, and PROFI3 funding, 311932) and European Research Council via ATM-GTP (742206). KRD acknowledges support by the SNF mobility grant P2EZP2_181599

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





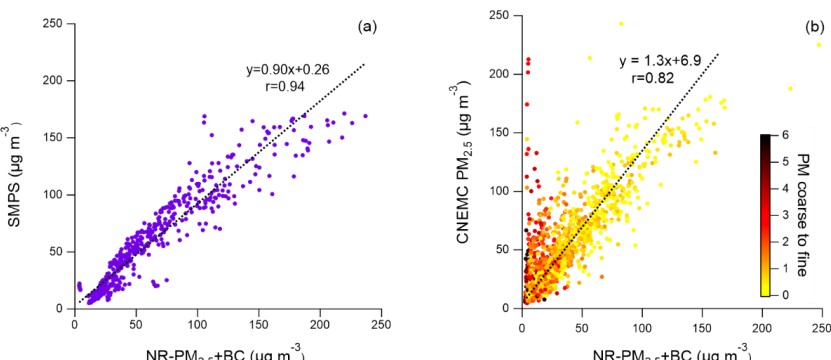


**Figure 1.** Scatter plot of (a) calculated PM mass concentration of SMPS and measured NR-PM$_{2.5}$+BC; (b) comparison between CNEMC PM$_{2.5}$ and NR-PM$_{2.5}$ + BC during the sampling period. PM$_{coarse\ to\ fine}$ is defined as the ratio of (PM$_{10}$ - PM$_{2.5}$)/PM$_{2.5}$ measured at CNEMC.

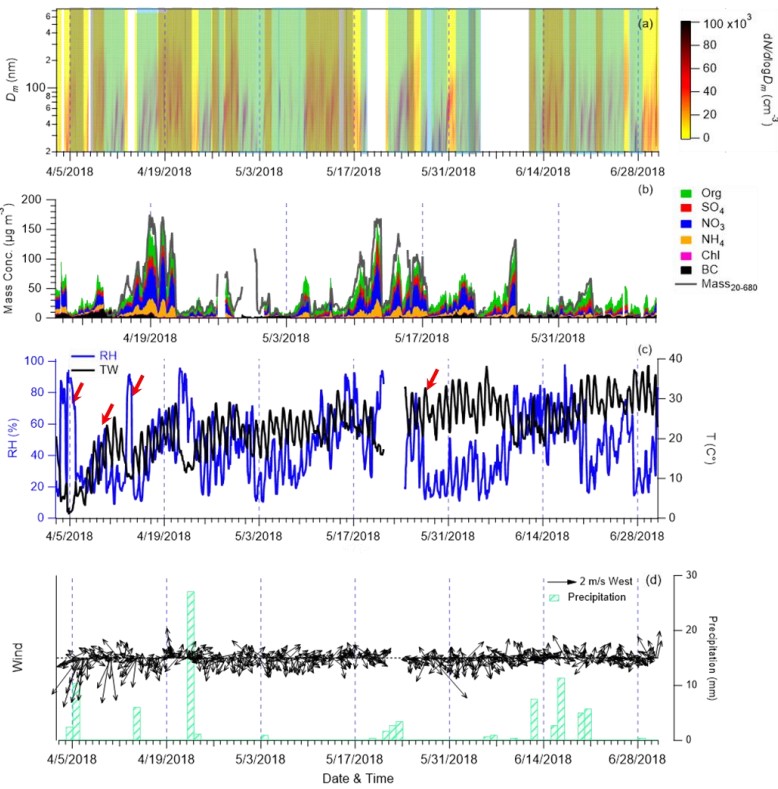

**Figure 2.** Temporal variation of (a) particle number size distribution, NPF and haze days are marked in blue and grey region, respectively; (b) mass concentrations of NR-PM$_{2.5}$ (including organics, sulphate, nitrate, ammonium and chloride) from ACSM and BC from AE-33; the comparison of hourly NR-PM$_{2.5}$+BC between calculated mass concentration from





SMPS; (c), RH (%), temperature (°C), red arrows indicate the arrivals of cold fronts; (d) 3-hour averaged wind direction, wind speed (m/s) and precipitation (precipitation data source: Beijing Nanyuan airport station).

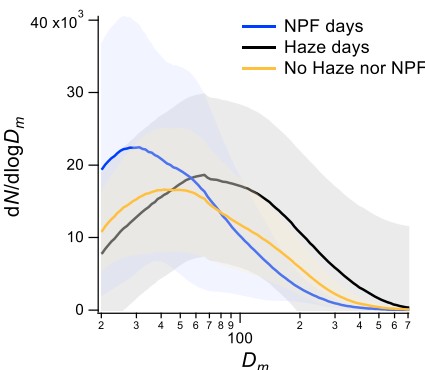


**Figure 3**. Average particle size distribution of NPF days, Haze days and No Haze nor NPF event days. The shadows represent one standard deviation. The average size distribution of PMF input are displayed in Figure S19. NPF days were excluded from the PMF analysis.

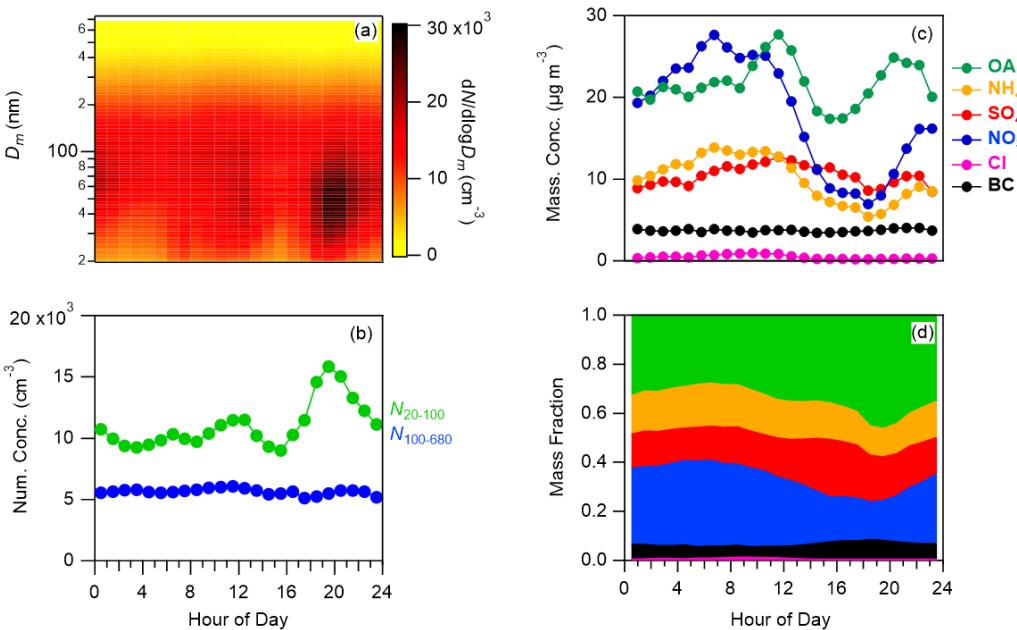

**Figure 4.** Average diurnal evolution of particle number size distribution during non-NPF days. (a) Particle size distribution, (b) number concentrations of particles in $N_{20\text{-}100}$ (20–100 nm) and $N_{100\text{-}680}$ (100–680 nm), (c) different component concentrations, and (d) mass fractions of different components. Average diurnal evolution of all days (NPF days are also included) are also presented in Figure S1.

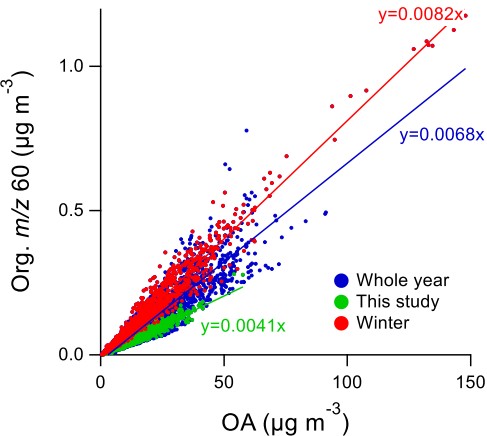

**Figure 5.** The ratio of *m/z* 60 in organics to total OA in our sampling period (April 6 to July 2, 2018), winter period (Dec 2018 to Feb 2019) and the whole year measurement (Feb 2018 to Jun 2019).

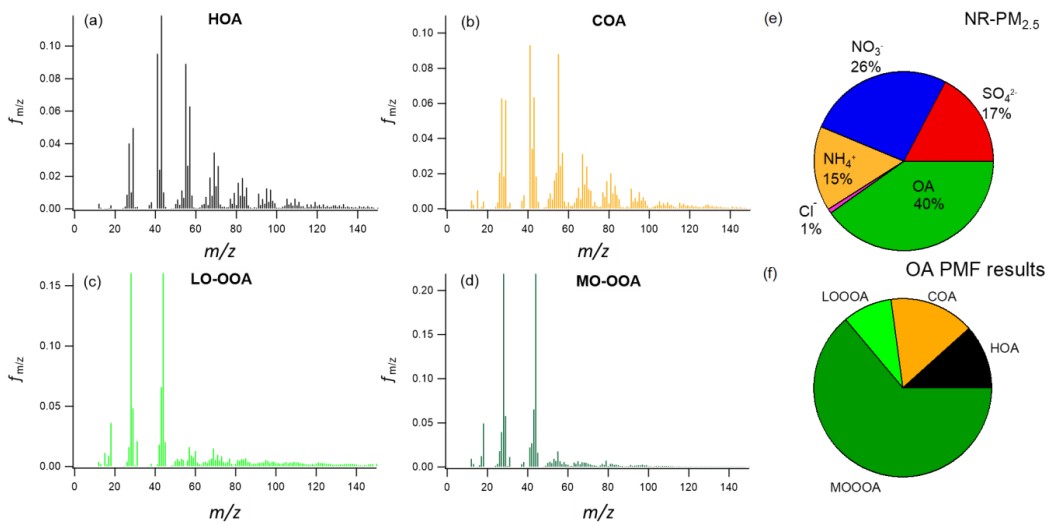

**Figure 6.** Profiles of organic aerosols of the factors from OA-PMF analysis, (a) HOA, (b) COA, (c) LO-OOA, (d) MO-
OOA; (e) the mass fractions of NR-PM$_{2.5}$, (f) the mass fractions of OA sources. HOA = hydrocarbon-like organic Aerosol, COA = cooking organic aerosol (COA), LO-OOA = less oxygenated organic aerosol (LO-OOA), and MO-OOA = more oxygenated organic aerosol (MO-OOA).



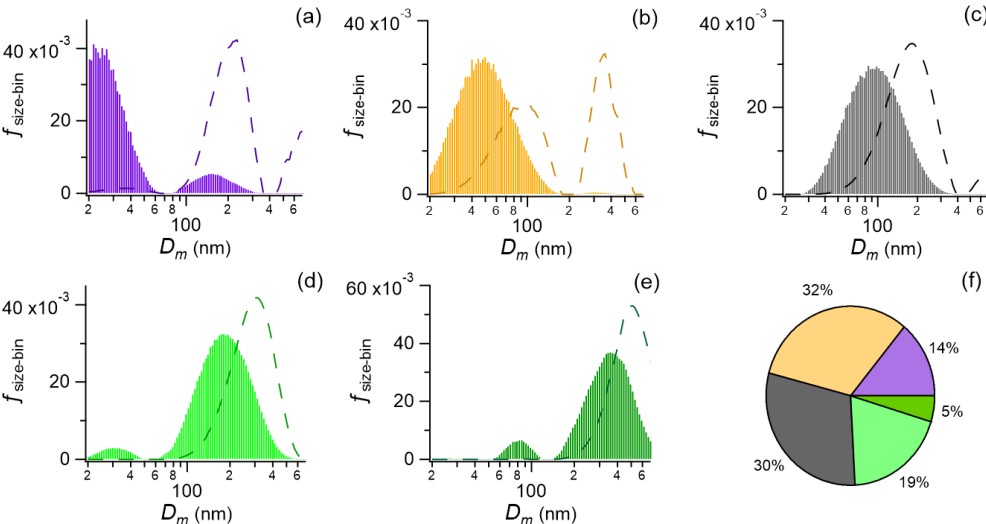

**Figure 7.** Number (shown as bars) and volume (shown as dashed lines) size distributions of the factors from Size-PMF
analysis and number fractions of total particle number concentrations, (a) Traffic-ultrafine, (b) Cooking-related, (c)
Traffic-fine, (d) Reg 1, (e) Reg 2, (f) number fractions of different factors. Traffic-ultrafine = ultrafine particles related
to traffic emissions, Traffic-fine = fine particles related to traffic emissions, Cooking-related = particles related to cooking
emissions, Reg 1 and Reg 2 = particles related to regional sources type 1 and 2, respectively.



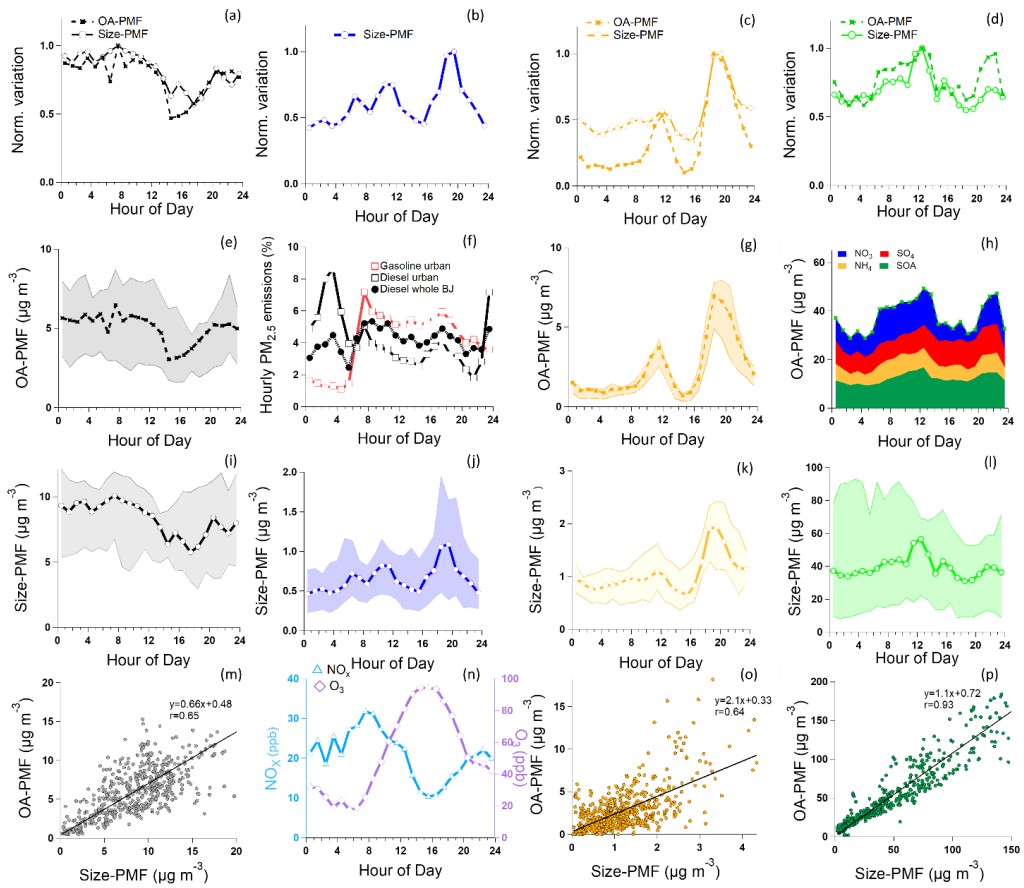

**Figure 8.** Normalized median diurnal variations (normalized by their highest median hourly value) of (a) HOA plus BC (black dash line) and Traffic-fine resolved from Size-PMF (black line); (b) Traffic-ultrafine from Size-PMF; (c) COA from ACSM (yellow dash line) and Cooking-related particles resolved from SMPS (yellow line); (d) Secondary aerosols (SIA+SOA) from ACSM (green dash line) and Regional-related from Size-PMF(green line); (e) diurnal patterns of HOA plus BC concentrations; (f) Simulated hourly $PM_{2.5}$ emissions from different traffic types (normalized by the highest median hourly values), red and black lines represented gasoline and diesel vehicles in urban Beijing, respectively, and black dash lines represented diesel vehicles in whole Beijing area; (g) diurnal patterns of COA concentrations; (h) diurnal patterns of SIA and SOA concentrations; diurnal variations of (i) Traffic-fine, (j) Traffic-ultrafine, (k) Cooking related (l) Regional-related; (m) comparison between HOA plus BC and Traffic-fine; (n) diurnal patterns of $NO_x$ (blue line) and $O_3$ (purple line); (o) comparison between COA and Cooking-related; (p) comparison between Secondary aerosols and Regional-related.

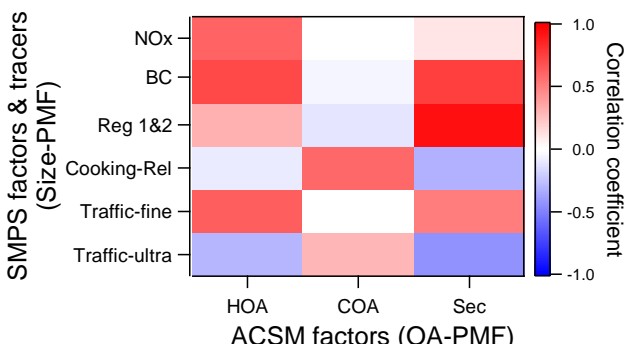

**Figure 9**. Pearson correlation coefficients (*r*) of ACSM versus SMPS factors, BC and NO$_x$ time series. Traffic-ultra = ultrafine particles related to traffic, Traffic-fine = fine particles related to traffic, Cooking Rel = particles related to
cooking activities, Reg 1&2 = sum of particles related to regional sources type 1 and 2, Sec = SIA+SOA,

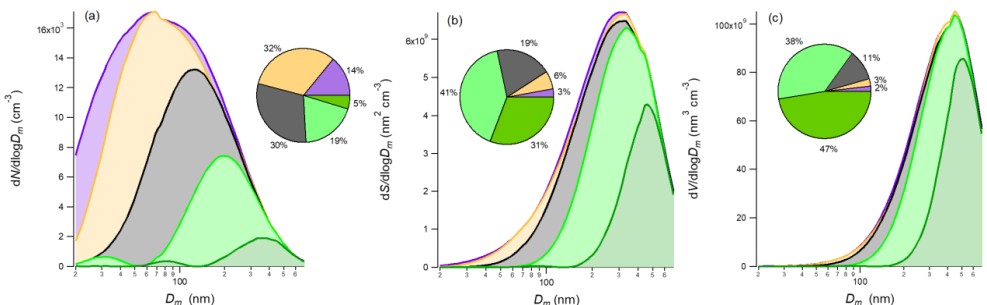

**Figure 10.** Profiles and fractions of (a) number, (b) surface area (c) and volume distributions from Size-PMF analysis.
Traffic-ultrafine = ultrafine particles related to traffic emissions (blue range), Traffic-fine = fine particles related to traffic
emissions (black range), Cooking Rel = particles related to cooking activities (yellow range), Reg 1 and Reg 2 = particles
related to regional sources type 1 (light green range) and 2 (dark green range), respectively.