# Peer review of "Size segregated particle number and mass concentrations from different emission sources in urban Beijing"

_Atmospheric Chemistry and Physics, 2020_

## Referee Comment (RC1) · Anonymous Referee #2 · 26 May 2020

This study combined particle number size distribution with chemical speciation source apportionment methods and compared their results in a comprehensive manner, which provides more detailed information about primary sources in Beijing for the sampling period. Overall, it is an interesting manuscript relevant to source apportionment of atmospheric particles in megacities. The authors also provided detailed supporting information on the method and its validations. In general, the paper is well written and fits well to the scope of ACP. I would like to recommend that the manuscript can be published on ACP after the following minor aspects are fully addressed.

1. Since Size-PMF was much less applied than OA-PMF, adding a summary table about the previous studies using this method in the introduction section (at least in the supplementary information) could help the potential readers better understand its

applications.

2. The noon peak of Traffic-ultrafine was explained by wind changes in the paper. However, it is also possible that some occasional NPFs and their following growth might also affect during this time of the day, even though NPFs days were fully excluded. The authors should also state this kind of possibility. Besides, it would be interesting to compare with the previous particle number PMF studies in Beijing.

3. Figure 2. The information of PNSD is not easy to follow when NPF and haze days are marked in blue and grey region. Maybe it could be clearer to use non-filled boxes. Besides, the legend of the subpanel (c) should be "T" rather than "TW".

4. Figure 4 a, c. Please also add time scale in those two sub-panels as the x-axis.

5. Figure 8, there are too many sub-panels providing similar information. The authors should make this figure easier to understand.

6. Line 113 – 118, it is the first time in the paper that the authors declared that there was no strong coal combustion and biomass burning emissions during their observation period. Yet, more explanations seem to be given at the section of 3.2. Some of the descriptions should be moved here.

7. Line 174 – 176, The CE of ACSM also depends on ambient RH variations. If a dryer was applied, it should be clearly stated in the Method section. If not, the authors should explain the possible influence of RH on CE and how to exclude it.

8. Line 230 – 231, cold front is a meteorological definition. If the authors declare that cold fronts are occurring, more evidence of meteorological parameters should be provided. In my point of view, those shape decreases of PM in summer were mostly caused by precipitation rather than cold fronts, which was shown in figure 2 (d).

9. Line 120, in the method section, the authors used the term of ToF-ACSM for the short of Time-of-Flight-Aerosol Chemical Speciation Monitor. Yet, in the following sections, the authors also used the term ACSM instead (such as in Line 167 and Line 176).

To make it different from Q-ACSM, it is better to always use the term of ToF-ACSM throughout the paper.

10. Line 369, the background for the nighttime Traffic-ultrafine type seems much higher than the simulated gasoline emissions. Except for diesel truck emissions and nighttime cluster formation listed, lower boundary layer during the nighttime would also be an important factor.
* * *

---

## Referee Comment (RC2) · Anonymous Referee #3 · 6 Aug 2020

In this study, by combined using both chemical fingerprints (OA-PMF) and particle size distribution (Size-PMF) analyses to resolve the particle mass and number contributions from various sources during the measurement period from April 6 to July 2, 2018, the authors have made efforts to better constrain the chemical and physical properties of primary organic aerosol in Beijing. They indicated that on days with no signs of new particle formation (NPF), primary emissions from traffic and cooking activities, contributed most to the particle number concentration below 100 nm while secondary mass formation dominated the total particle mass concentration. Overall, this manuscript is well organized and present with new interesting results to readers and policymakers, which benefit for better understanding of the sources of PM2.5 in megacities like Beijing. Thus, this reviewer recommend it be accepted for publication

in ACP after made several minor revisions.

1. Title: I suggest the title should be "Size segregated particle number and mass concentrations in urban Beijing" because both the number and mass concentrations of PM2.5 sampled by ACSM and SMPS. 2. Line 306-308: you only mentioned four factors rather than five here. Please check. 3. If possible, some additional discussion on comparison with former studies in Beijing with ACMS is added for better tracking the change of emission sources in Beijing since 2013.

---

## Author Comment (AC1)

**Responses to the reviewers' comments for manuscript**

**Size segregated particle number and mass emissions in urban Beijing**

**Title revised to "Size segregated particle number and mass concentrations of emission sources in urban Beijing" according to Reviewer #2's comment**

Jing Cai[1,2], Biwu Chu[1,2,3,4], Lei Yao[2], Chao Yan[1,2], Liine M. Heikkinen[1,2], Feixue Zheng[1], Chang Li[1], Xiaolong Fan[1], Shaojun Zhang[5], Daoyuan Yang[5], Yonghong Wang[2], Tom V. Kokkonen[1,2], Tommy Chan[1,2], Ying Zhou[1], Lubna Dada[1,2], Yongchun Liu[1], Hong He[3,4], Pauli Paasonen[1,2], Joni T. Kujansuu[1,2], Tuukka Petäjä[1,2], Claudia Mohr[6], Juha Kangasluoma[1,2], Federico Bianchi[1,2], Yele Sun[7], Philip L. Croteau[8], Douglas R. Worsnop[2,8], Veli-Matti Kerminen[1,2], Wei Du[1,2]*, Markku Kulmala[1,2]*, Kaspar R. Daellenbach[1,2]*

[1] Aerosol and Haze Laboratory, Beijing Advanced Innovation Center for Soft Matter Science and Engineering, Beijing University of Chemical Technology, Beijing, 100029, China

[2] Institute for Atmospheric and Earth System Research, Faculty of Science, University of Helsinki, Helsinki, 00014, Finland

[3] Center for Excellence in Regional Atmospheric Environment, Institute of Urban Environment, Chinese Academy of Sciences, Xiamen, 361021, China

[4] State Key Joint Laboratory of Environment Simulation and Pollution Control, Research Center for Eco-Environmental Sciences, Chinese Academy of Sciences, Beijing, 100085, China

[5] School of Environment, Tsinghua University, Beijing, 100084, China

[6] Department of Environmental Science, Stockholm University, Stockholm, 11418, Sweden

[7] State Key Laboratory of Atmospheric Boundary Layer Physics and Atmospheric Chemistry, Institute of Atmospheric Physics, Chinese Academy of Sciences, Beijing, 100029, China

[8] Aerodyne Research, Inc., Billerica, MA, 01821, USA

*Correspondence to:* Wei Du (wei.du@helsinki.fi), Markku Kulmala (markku.kulmala@helsinki.fi) and Kaspar R. Daellenbach (kaspar.daellenbach@gmail.com)

We thank all the reviewers for their evaluation of the manuscript, and for their valuable and constructive feedback. Referee comments are given in black italics and the replies to the individual comments are directly added below them in regular black typeset. Changes made in the manuscript are in blue and marked with underlines. Page and line numbers refer to the original manuscript text.

**Reviewer #1**

*This study combined particle number size distribution with chemical speciation source apportionment methods and compared their results in a comprehensive manner, which provides more detailed information about primary sources in Beijing for the sampling period. Overall, it is an interesting manuscript relevant to source apportionment of atmospheric particles in megacities. The authors also provided detailed supporting information on the method and its validations. In general, the paper is well written and fits well to the scope of ACP. I would like to recommend that the manuscript can be published on ACP after the following minor aspects are fully addressed.*

Response: We thank the reviewer for the comments and their recommendation to eventually publish our manuscript.

Comments:

*1. Since Size-PMF was much less applied than OA-PMF, adding a summary table about the previous studies using this method in the introduction section (at least in the supplementary information) could help the potential readers better understand its applications.*

Response: We have added a summary table (Table S1) about the application of Size-PMF from previous literature in the revised supplementary information as suggested. Also, we present Table S1 here.

Table S1. Sources identified by Size-PMF method

| Sampling site | Sampling year | Source types | Sampling equipment | Reference |
|---|---|---|---|---|
| Augsburg, Germany | Winter 2007/07 | Re-suspended dust, fresh/aged traffic, combustion, long-range transported dust, nucleation, secondary aerosols | UDMA-UCPC/DMA-CPC/APS | Gu et al. (2011) |
| Barcelona, Spain | Jan 2013–Dec 2016 | Photonucleation, traffic nucleation, urban, secondary | SMPS | Rivas et al. (2020) |
| | Nov 2003–Dec 2004 | Road traffic, mineral dust, industry, sea spray, photonucleation, regional, combustion | DMPS | Pey et al. (2009) |
| Beijing, China | August 2008 | Local/distant traffic, secondary nitrate, combustion | TDMPS | Wang et al. (2013) |
| | Aug–Sep, 2015 | Nucleation, local primary emissions (e.g., cooking and traffic emissions), secondary | SMPS | Du et al. (2017) |
| Helsinki, Finland | Feb 2015–Aug 2017, Jan 2007–Dec 2016 | Photonucleation, traffic nucleation, fresh traffic, urban, biogenic, secondary | DMPS | Rivas et al. (2020) |

| | | | | |
|---|---|---|---|---|
| | Jan 2010–Dec 2016, Mar 2014–Dec 2016 | Photonucleation, traffic nucleation, fresh traffic, urban, secondary | SMPS | Rivas et al. (2020) |
| London, UK | Oct–Nov, 2007 | Road emissions (vehicle exhaust, brake dust, resuspension), Urban background (accumulation mode, suburban traffic, solid fuel/nitrate, regional, cooking, regional) | SMPS/APS | Harrison et al. (2011) |
| Pittsburgh, US | Jun–Aug, 2001 | Local/distant traffic, secondary nitrate, regional transport, combustion | Nano-SMPS/SMPS/APS | Zhou et al. (2004) |
| Rochester, NY, US | Dec 2004–Nov 2005 | Nucleation, traffic, industry, heating, secondary nitrate, secondary sulfate, regional transport | SMPS | Ogulei et al. (2007) |
| Zurich, Switzerland | Dec 2010–Oct 2014 | Photonucleation, traffic nucleation, fresh traffic, urban, secondary | SMPS | Rivas et al. (2020) |

*2. The noon peak of Traffic-ultrafine was explained by wind changes in the paper. However, it is also possible that some occasional NPFs and their following growth might also affect during this time of the day, even though NPFs days were fully excluded. The authors should also state this kind of possibility. Besides, it would be interesting to compare with the previous particle number PMF studies in Beijing.*

Response: As the reviewer states, we excluded days during which new particle formation (NPF) events occurred and only considered days during which no NPF days occurred (non-NPF days). Since we defined NPF events based on particle size distributions starting at 2nm (the size range in which new particles form and start to grow) (NAIS and SMPS) and only consider size bins starting at 20nm (SMPS), the impact of NPF on our results can be considered small. In addition, a previous Size-PMF study conducted in Beijing observed a similar noon peak from traffic emissions (Wang et al., 2013).

However, as suggested by the reviewer, a minor contribution from newly formed particles and their following growth into the observed size range cannot be completely excluded. Therefore, we state in the text that we minimize the impact of NPF:

"In this study, we minimized the effect of NPF on the Size-PMF results by excluding NPF days based on an evaluation starting from2nm particle concentrations from the NAIS."

In response to the reviewer's comment, we added the following content to the manuscript:

Line 370: "This causes the measurement location to be strongly affected by the main intersections and arterial roads upwind (Figure S15). However, considering the absence of strong nucleation mode particles burst this peak was far more likely to originate from the primary emissions such as gasoline vehicle emission, which is supported by a shoulder of the $NO_x$ peak at the same time of day that was still observed even though $NO_x$ is depleted by increasing concentrations of $O_3$."

*3. Figure 2. The information of PNSD is not easy to follow when NPF and haze days are marked in blue and grey region. Maybe it could be clearer to use non-filled boxes. Besides, the legend of the subpanel (c) should be "T" rather than "TW".*

Response: We thank the reviewer for their suggestions. We have revised Figure 2 as suggested and corrected the legend in subpanel (c). In addition, to reduce possible misunderstandings, we have also changed the x-axis scale of Figure 2 (b), making it the same as the other panels of the same figure. The revised Figure 2 is as follows:

[Figure]

**Figure 2.** Temporal variation of (a) particle number size distribution, NPF and haze days are marked with blue and grey boxes, respectively; (b) mass concentrations of NR-PM$_{2.5}$ (including organics, sulphate, nitrate, ammonium and chloride) from ACSM and BC from AE-33; the comparison of hourly NR-PM$_{2.5}$+BC between calculated mass concentration from SMPS;

(c), RH (%), temperature (°C), red arrows indicate the arrivals of cold fronts; (d) 3-hour averaged wind direction, wind speed (m/s) and precipitation (precipitation data source: Beijing Nanyuan airport station).

*4. Figure 4 a, c. Please also add time scale in those two sub-panels as the x-axis.*

Response: We thank the reviewer's suggestions and the Figure 4 has been revised as follows:

[Figure]

**Figure 4.** Average diurnal evolution of particle number size distribution during non-NPF days. (a) Particle size distribution, (b) number concentrations of particles in $N_{20\text{-}100}$ (20–100 nm) and $N_{100\text{-}680}$ (100–680 nm), (c) different component concentrations, and (d) mass fractions of different components. Average diurnal evolution of all days (NPF days are also included) are also presented in Figure S1.

*5. Figure 8, there are too many sub-panels providing similar information. The authors should make this figure easier to understand.*

Response: In response to the reviewers comment and to better exhibit the result, we deleted the row of subpanels of the original Figure 8(a) to (d) and moved them to the supplementary information (Figure S20). The original and revised Figure 8 are shown below:

[Figure]

Original Figure 8. Normalized median diurnal variations (normalized by their highest median hourly value) of (a) HOA plus BC (black dash line) and Traffic-fine resolved from Size-PMF (black line); (b) Traffic-ultrafine from Size-PMF; (c) COA from ACSM (yellow dash line) and Cooking-related particles resolved from SMPS (yellow line); (d) Secondary aerosols (SIA+SOA) from ACSM (green dash line) and Regional-related from Size-PMF(green line); (e) diurnal patterns of HOA plus BC concentrations; (f) Simulated hourly PM2.5 emissions from different traffic types (normalized by the highest median hourly values), red and black lines represented gasoline and diesel vehicles in urban Beijing, respectively, and black dash lines represented diesel vehicles in whole Beijing area; (g) diurnal patterns of COA concentrations; (h) diurnal patterns of SIA and SOA concentrations; diurnal variations of (i) Traffic-fine, (j) Traffic-ultrafine, (k) Cooking related (l) Regional-related; (m) comparison between HOA plus BC and Traffic-fine; (n) diurnal patterns of NOx (blue line) and $O_3$ (purple line); (o) comparison between COA and Cooking-related; (p) comparison between Secondary aerosols and Regional-related.

[Figure]

Revised Figure 8. Median diurnal patterns of (a) Traffic-ultrafine resolved from Size-PMF, (b) Traffic-fine from Size-PMF, (c) Cooking-related particles resolved from SMPS, (d) Regional-related from SMPS, (e) Simulated hourly variation of $PM_{2.5}$ emissions from different traffic types. For each traffic type, the proportions are calculated from hourly emissions divided by the whole day emissions of its type; red and black lines represent gasoline and diesel vehicles in urban Beijing, respectively, and black dash lines represent diesel vehicles in whole Beijing area, (f) HOA plus BC from ToF-ACSM and aethalometer, (g) COA from ToF-ACSM, (h) Secondary from ToF-ACSM; (i) $NO_x$ and $O_3$, blue line represented $NO_x$, and purple line represented $O_3$, (j) comparison between HOA plus BC and Traffic-fine, (k) comparison between COA and Cooking-related, (l) comparison between Secondary species from ACSM and Regional from SMPS. Shaded areas are 25th and 75th percentile.

*6. Line 113 – 118, it is the first time in the paper that the authors declared that there was no strong coal combustion and biomass burning emissions during their observation period. Yet, more explanations seem to be given at the section of 3.2. Some of the descriptions should be moved here.*

Response: In response to the reviewer's comment we reformulated the respective paragraph and refer to literature and our own analysis where needed: "Coal and biomass burning from the residential sectors are more important during winter in Beijing and the North China Plain (Hu et al., 2017;Sun et al., 2018). In addition, the transition in energy consumption from coal burning to natural gas and electricity in urban Beijing took place from the year of 2009 to 2017, which led to a decrease in the proportion of coal to total primary energy consumption from 43% in 2007 to less than 20% in 2015 (Zhang et al., 2018). The effects of residential coal combustion and biomass burning

were not strong during our sampling period, which is supported by the chemical component measurements. More supporting information of the absence of the residential combustion emissions during the sampling period is provided in the section of 3.2 and SI (Figure S5 and S12)"

*7. Line 174 – 176, The CE of ACSM also depends on ambient RH variations. If a dryer was applied, it should be clearly stated in the Method section. If not, the authors should explain the possible influence of RH on CE and how to exclude it.*

Response: We added the following information in the method section:

Line 121– 123 "A $PM_{2.5}$ cyclone was deployed on the rooftop with a flow rate of 3 L.min$^{-1}$ and connected to the ToF-ACSM by a 3-m stainless steel tube through a Nafion dryer (Perma-Pure, MD-700-24F-3)."

*8. Line 230 – 231, cold front is a meteorological definition. If the authors declare that cold fronts are occurring, more evidence of meteorological parameters should be provided. In my point of view, those shape decreases of PM in summer were mostly caused by precipitation rather than cold fronts, which was shown in figure 2 (d).*

Response: In response to the reviewer's concern, we changed the statement in the revised manuscript.

Original statement: "The sharp decreases of the particle number and mass concentrations in the sampling period can be explained by the arrivals of cold fronts and precipitation. The cold fronts are indicated by a wind shift from the South to the North and significant increase of wind speed shown in Figure 2 d."

Adapted statement: "The sharp decreases of the particle number and mass concentrations in the sampling period can be mainly explained by precipitation. Besides, the strong north wind could also largely decrease the fine particulate matter concentrations such as May 27$^{th}$ shown in Figure 2 (d)."

*9. Line 120, in the method section, the authors used the term of ToF-ACSM for the short of Time-of-Flight-Aerosol Chemical Speciation Monitor. Yet, in the following sections, the authors also used the term ACSM instead (such as in Line 167 and Line 176). To make it different from Q-ACSM, it is better to always use the term of ToF-ACSM throughout the paper.*

Response: We have revised the term of ACSM to ToF-ACSM throughout the manuscript as suggested.

*10. Line 369, the background for the nighttime Traffic-ultrafine type seems much higher than the simulated gasoline emissions. Except for diesel truck emissions and nighttime cluster formation listed, lower boundary layer during the nighttime would also be an important factor.*

Response: We now have revised the statement from the original manuscript as follows:

From: "The background late night particle concentrations can be attributed to nighttime cluster formation, or to sub-30 nm particle emissions from HDV emissions (Song et al., 2013;Wehner et al., 2009)."

To: "The background late-night particle concentrations can be attributed to nighttime cluster formation, or to sub-30 nm particle emissions from HDV emissions (Song et al., 2013;Wehner et al., 2009). In addition, the lower boundary layer during night increases particle concentrations."

**Reviewer #2**

*In this study, by combined using both chemical fingerprints (OA-PMF) and particle size distribution (Size-PMF) analyses to resolve the particle mass and number contributions from various sources during the measurement period from April 6 to July 2, 2018, the authors have made efforts to better constrain the chemical and physical properties of primary organic aerosol in Beijing. They indicated that on days with no signs of new particle formation (NPF), primary emissions from traffic and cooking activities, contributed most to the particle number concentration below 100 nm while secondary mass formation dominated the total particle mass concentration. Overall, this manuscript is well organized and present with new interesting results to readers and policymakers, which benefit for better understanding of the sources of PM$_{2.5}$ in megacities like Beijing. Thus, this reviewer recommend it be accepted for publication in ACP after made several minor revisions.*

Response: We thank the reviewer for the comments and their recommendation to eventually publish our manuscript. The revisions according to the comments are listed below.

*1. Title: I suggest the title should be "Size segregated particle number and mass concentrations in urban Beijing" because both the number and mass concentrations of PM$_{2.5}$ sampled by ACSM and SMPS.*

Response: In response to the reviewer's comment, we adapted the manuscript's title. As the reviewer points out, in this study we examine the number as well as the mass concentration instead of focusing on one or the other. Since we focused on the emission sources by performing source apportionment analyses of the particles mass as well as the number, we chose a title that reflects that:

Size segregated particle number and mass concentrations from different emission sources in urban Beijing

*2. Line 306-308: you only mentioned four factors rather than five here. Please check.*

Response: In the original version, the five factors from Line 306-308 were "Traffic-fine particles (Traffic-fine), Traffic-ultrafine particles (Traffic-ultrafine), cooking activity related particles (Cooking-related) and Regional particles". Here the Regional particles included two types of Regional particles (Regional 1& Regional 2) with different sizes. To make it clear, we have revised the sentence as follows:

"Traffic-fine particles (Traffic-fine), Traffic-ultrafine particles (Traffic-ultrafine), cooking activity related particles (Cooking-related) and two kinds of Regional particles (Regional 1 & 2)"

*3. If possible, some additional discussion on comparison with former studies in Beijing with ACMS is added for better tracking the change of emission sources in Beijing since 2013.*

Response: We thank the reviewer for this suggestion and have added more information from previous ACSM/AMS studies in the revised manuscript. Yet, from previous studies, the sources of fine particles were reported to be varied among different seasons, especially for heating/non-heating seasons. Besides, previous ACSM or AMS studies in Beijing typically applied an aerodynamic lens transmitting PM$_1$ only rather than PM$_{2.5}$ like in this study,

making the direct comparison even more difficult, especially for the absolute concentrations. Thus, we added a discussion on previous studies in the main text as suggested, yet, also emphasize the potential influence of different sampling sites and PM size cut on the comparisons. In the supplementary information, we have added information on different OA component fractions from previous studies conducted during non-heating seasons from the years 2008–2018 (Table S2, shown below) as suggested.

"The contribution of aerosol components to NR-PM$_{2.5}$ and the chemical fingerprints from OA-PMF are displayed in Figure 6. Generally, the source types and contributions exhibited a large fraction of OOA, consistent, with those from previous studies conducted in the same seasons in urban Beijing (Hu et al., 2017;Sun et al., 2018). We observe a slightly higher contribution of SOA to OA (73% for Apr-July 2018 in this study) than what was reported in literature for the early 2000s (65%–68%) (Table S2). The decreased contribution of POA to OA compared to the early 2000s is likely related to the implementation of emission controls for the recent years in Beijing. Yet, it should also be noted that different factors might affect the comparison, such as sampling location, the uncertainties in source apportionment, as well as particle size cuts."

Table S2. The mass fractions of resolved OA component in Beijing conducted in non-heating season period

| Sampling time | Source | Prop. to OA | Size | Reference |
|---|---|---|---|---|
| | HOA | 18% | | |
| | COA | 24% | | |
| Jul–Sep 2008 | OOA1 | 34% | PM$_1$ | Huang et al. (2010) |
| | OOA2 | 35% | | |
| | HOA | 13% | | |
| | COA | 21% | | |
| Aug–Sep 2011 | LO-OOA | 28% | PM$_1$ | Hu et al. (2016) |
| | MO-OOA | 37% | | |
| | HOA | 11% | | |
| | COA | 20% | | |
| Jul–Aug 2012 | LO-OOA | 43% | PM$_1$ | Hu et al. (2017) |
| | MO-OOA | 26% | | |

| | | | | |
|---|---|---|---|---|
| | FFOA | 9% | | |
| | COA | 13% | | |
| May 2013 | BBOA | 6% | PM$_1$ | Sun et al. (2018) |
| | LO-OOA | 14% | | |
| | MO-OOA | 58% | | |
| | FFOA | 6% | | |
| | COA | 12% | | |
| Oct 2014 | BBOA | 10% | PM$_1$ | Zhou et al. (2018) |
| | LO-OOA | 15% | | |
| | OOA | 54% | | |
| | HOA | 8% | | |
| | COA | 18% | | |
| Jul–Aug 2015 | ISOOA (isoprene-oxidized OA) | 5% | PM$_1$ | Duan et al. (2020) |
| | OOA | 68% | | |
| | HOA | 13% | | |
| | COA | 15% | | |
| Jun 2017 | LO-OOA | 45% | PM$_1$ | Xu et al. (2019) |
| | MO-OOA | 27% | | |
| | HOA | 11% | | |
| | COA | 24% | | |
| May–Jun 2018 | LO-OOA | 39% | PM$_1$ | Xu et al. (2019) |
| | MO-OOA | 26% | | |
| | HOA | 12% | | |
| | COA | 15% | | |
| Apr–July 2018 | LO-OOA | 9% | PM$_{2.5}$ | This study |
| | MO-OOA | 64% | | |

**Reference**

Du, W., Zhao, J., Wang, Y. Y., Zhang, Y. J., Wang, Q. Q., Xu, W. Q., Chen, C., Han, T. T., Zhang, F., Li, Z. Q., Fu, P. Q., Li, J., Wang, Z. F., and Sun, Y. L.: Simultaneous measurements of particle number size distributions at ground level and 260m on a meteorological tower in urban Beijing, China, Atmospheric Chemistry and Physics, 17, 6797-6811, 10.5194/acp-17-6797-2017, 2017.

Duan, J., Huang, R.-J., Li, Y., Chen, Q., Zheng, Y., Chen, Y., Lin, C., Ni, H., Wang, M., Ovadnevaite, J., Ceburnis, D., Chen, C., Worsnop, D. R., Hoffmann, T., O'Dowd, C., and Cao, J.: Summertime and wintertime atmospheric processes of secondary aerosol in Beijing, Atmospheric Chemistry and Physics, 20, 3793-3807, 10.5194/acp-20-3793-2020, 2020.

Gu, J., Pitz, M., Schnelle-Kreis, J., Diemer, J., Reller, A., Zimmermann, R., Soentgen, J., Stoelzel, M., Wichmann, H. E., Peters, A., and Cyrys, J.: Source apportionment of ambient particles: Comparison of positive matrix factorization analysis applied to particle size distribution and chemical composition data, Atmospheric Environment, 45, 1849-1857, 10.1016/j.atmosenv.2011.01.009, 2011.

Harrison, R. M., Beddows, D. C., and Dall'Osto, M.: PMF analysis of wide-range particle size spectra collected on a major highway, Environ Sci Technol, 45, 5522-5528, 10.1021/es2006622, 2011.

Hu, W., Hu, M., Hu, W. W., Zheng, J., Chen, C., Wu, Y. S., and Guo, S.: Seasonal variations in high time-resolved chemical compositions, sources, and evolution of atmospheric submicron aerosols in the megacity Beijing, Atmospheric Chemistry and Physics, 17, 9979-10000, 10.5194/acp-17-9979-2017, 2017.

Hu, W. W., Hu, M., Hu, W., Jimenez, J. L., Yuan, B., Chen, W. T., Wang, M., Wu, Y. S., Chen, C., Wang, Z. B., Peng, J. F., Zeng, L. M., and Shao, M.: Chemical composition, sources, and aging process of submicron aerosols in Beijing: Contrast between summer and winter, Journal of Geophysical Research-Atmospheres, 121, 1955-1977, 10.1002/2015jd024020, 2016.

Huang, X. F., He, L. Y., Hu, M., Canagaratna, M. R., Sun, Y., Zhang, Q., Zhu, T., Xue, L., Zeng, L. W., Liu, X. G., Zhang, Y. H., Jayne, J. T., Ng, N. L., and Worsnop, D. R.: Highly time-resolved chemical characterization of atmospheric submicron particles during 2008 Beijing Olympic Games using an Aerodyne High-Resolution Aerosol Mass Spectrometer, Atmospheric Chemistry and Physics, 10, 8933-8945, 10.5194/acp-10-8933-2010, 2010.

Ogulei, D., Hopke, P. K., Chalupa, D. C., and Utell, M. J.: Modeling Source Contributions to Submicron Particle Number Concentrations Measured in Rochester, New York, Aerosol Science and Technology, 41, 179-201, 10.1080/02786820601116012, 2007.

Pey, J., Querol, X., Alastuey, A., Rodríguez, S., Putaud, J. P., and Van Dingenen, R.: Source apportionment of urban fine and ultra-fine particle number concentration in a Western Mediterranean city, Atmospheric Environment, 43, 4407-4415, 10.1016/j.atmosenv.2009.05.024, 2009.

Rivas, I., Beddows, D. C. S., Amato, F., Green, D. C., Jarvi, L., Hueglin, C., Reche, C., Timonen, H., Fuller, G. W., Niemi, J. V., Perez, N., Aurela, M., Hopke, P. K., Alastuey, A., Kulmala, M., Harrison, R. M., Querol, X., and Kelly, F. J.: Source apportionment of particle number size distribution in urban background and traffic stations in four European cities, Environ Int, 135, 105345, 10.1016/j.envint.2019.105345, 2020.

Song, S., Wu, Y., Xu, J., Ohara, T., Hasegawa, S., Li, J., Yang, L., and Hao, J.: Black carbon at a roadside site in Beijing: Temporal variations and relationships with carbon monoxide and particle number size distribution, Atmospheric Environment, 77, 213-221, 10.1016/j.atmosenv.2013.04.055, 2013.

Sun, Y., Xu, W., Zhang, Q., Jiang, Q., Canonaco, F., Prévôt, A. S. H., Fu, P., Li, J., Jayne, J., Worsnop, D. R., and Wang, Z.: Source apportionment of organic aerosol from 2-year highly time-resolved measurements by an aerosol chemical speciation monitor in Beijing, China, Atmospheric Chemistry and Physics, 18, 8469-8489, 10.5194/acp-18-8469-2018, 2018.

Sun, Y., He, Y., Kuang, Y., Xu, W., Song, S., Ma, N., Tao, J., Cheng, P., Wu, C., Su, H., Cheng, Y., Xie, C., Chen, C., Lei, L., Qiu, Y., Fu, P., Croteau, P., and Worsnop, D. R.: Chemical Differences Between PM1 and PM 2.5 in Highly Polluted Environment and Implications in Air Pollution Studies, Geophysical Research Letters, 47, 10.1029/2019gl086288, 2020.

Wang, Z. B., Hu, M., Wu, Z. J., Yue, D. L., He, L. Y., Huang, X. F., Liu, X. G., and Wiedensohler, A.: Long-term measurements of particle number size distributions and the relationships with air mass history and source apportionment in the summer of Beijing, Atmospheric Chemistry and Physics, 13, 10159-10170, 10.5194/acp-13-10159-2013, 2013.

Wehner, B., Uhrner, U., von Löwis, S., Zallinger, M., and Wiedensohler, A.: Aerosol number size distributions within the exhaust plume of a diesel and a gasoline passenger car under on-road conditions and determination of emission factors, Atmospheric Environment, 43, 1235-1245, 10.1016/j.atmosenv.2008.11.023, 2009.

Xu, W., Xie, C., Karnezi, E., Zhang, Q., Wang, J., Pandis, S. N., Ge, X., Zhang, J., An, J., Wang, Q., Zhao, J., Du, W., Qiu, Y., Zhou, W., He, Y., Li, Y., Li, J., Fu, P., Wang, Z., Worsnop, D. R., and Sun, Y.: Summertime aerosol volatility measurements in Beijing, China, Atmospheric Chemistry and Physics, 19, 10205-10216, 10.5194/acp-19-10205-2019, 2019.

Zhang, P., Zhang, L., Tian, X., Hao, Y., and Wang, C.: Urban energy transition in China: Insights from trends, socioeconomic drivers, and environmental impacts of Beijing, Energy Policy, 117, 173-183, 10.1016/j.enpol.2018.02.039, 2018.

Zhou, L., Kim, E., Hopke, P. K., Stanier, C. O., and Pandis, S.: Advanced Factor Analysis on Pittsburgh Particle Size-Distribution Data Special Issue ofAerosol Science and Technologyon Findings from the Fine Particulate Matter Supersites Program, Aerosol Science and Technology, 38, 118-132, 10.1080/02786820390229589, 2004.

Zhou, W., Wang, Q., Zhao, X., Xu, W., Chen, C., Du, W., Zhao, J., Canonaco, F., Prévôt, A. S. H., Fu, P., Wang, Z., Worsnop, D. R., and Sun, Y.: Characterization and source apportionment of organic aerosol at 260 m on a meteorological tower in Beijing, China, Atmospheric Chemistry and Physics, 18, 3951-3968, 10.5194/acp-18-3951-2018, 2018.